# Developmental Differences on the Internal Reproductive Systems between the Prediapause and Prereproductive *Riptortus pedestris* Adults

**DOI:** 10.3390/insects11060347

**Published:** 2020-06-04

**Authors:** Abdul Hafeez, Bei Li, Wen Liu, Muhammad Nauman Atiq, Xiao-Ping Wang

**Affiliations:** Hubei Key Laboratory of Insect Resources Utilization and Sustainable Pest Management, College of Plant Science and Technology, Huazhong Agricultural University, Wuhan 430070, China; abdulhafeez861@gmail.com (A.H.); lb112358@webmail.hzau.edu.cn (B.L.); liuwen@mail.hzau.edu.cn (W.L.); mian.nauman911@gmail.com (M.N.A.)

**Keywords:** *Riptortus pedestris*, reproductive diapause, photoperiod, internal reproductive organ, morphology

## Abstract

*Riptortus pedestris* (Heteroptera: Alydidae), an important crop pest, is capable of entering reproductive adult diapause under short-day photoperiods. Though the physiological aspects of adult diapause have been well studied in this species, little is known about its morphological development. In the present study, the adult females are discriminated as prediapause and prereproductive based on the absence and presence of mature oocytes in ovarioles, respectively. We also measured the morphological development of vitellarium and lateral oviduct in females, and the accessory gland, ejaculatory duct, and testes in males of both prereproductive and prediapause adults. Our results revealed that there is a clear significant difference in the reproductive development of prediapause and prereproductive insects. Moreover, the internal morphology of reproductive organs was suppressed in prediapausebugs compared to prereproductive bugs, and the insects developedthe reproductive parts as newly emerged adults. The above findings provide basic knowledge on the characterization of diapause and reproductive *R. pedestris* adults, which would be applicable to molecular investigations.

## 1. Introduction 

Reproductive development is a universal event that takes place under favorable environmental conditions [1]. In response to change in environmental conditions, insects undergo an alternative period of suppression in reproductive development rather than developing into a reproductive one [2]. These conditions would be achange in temperature or photoperiod, shortage of food, orthe absence of oviposition sites [3]. Tosynchronizewith the environmental conditions, insects adopt several strategies, including altering their behavioral response (migration, reconstructingamicrohabitat, or digging into the soil), suppression in hormonal regulation, or undergoing diapause. Among these strategies, reproductive diapause is a very common state adopted by many species of insects, including Coleoptera, Blattaria, Diptera, Hemiptera, Hymenoptera, Lepidoptera, and Orthoptera [4]. During adult diapause, insects undergo a period of arrestment by reducing the regulation of hormones and suppressing reproductive development [3,5]. This kind of suppression program does not proceed abruptly; rather, it is induced by a series of systematic events that occur in theprediapausephaseof an insect [6]. The prediapause phase of development can be distinguished by two successive phases, known as theinduction phase and the preparation phase. These phases of prediapausewould be separated or overlapped depending upon the insect species [6,7]. Generally, the initiation of adult diapause can easily be characterized by thechange in the color orshape of some morphological structuresor by inhibiting the reproductive development of an adult. In addition to reproductive development, some specific responses of the behaviorcan also be recognized during the induction phase of adult diapause, as the newly emerged adults of *Colaphellus bowringi* start intense feeding before theydig into the soil and then undergo diapause [8,9]. However, the preparation phase of diapause is the critical phase in the life cycle of diapausing insects in which several major changes occur, including the accumulation of nutrients, the inhibition of reproduction, and the suppression of hormonal secretion, which decide the survival of an insect during thestress period by the environment [5,10].

The bean bug, *Riptortus pedestris*, is a polyphagous insect that hasmore than one generation in a year depending upon the environmental conditions. This species has overwintering adult diapause under leaf litters [11,12]. Physiological studies have shown that both nymph and adults of *R. pedestris* are sensitive to environmental conditions, but the fourth and fifth instars of nymphs are more sensitive than the adults to short-day photoperiods [13]. The females of *R. pedestris* enter reproductive diapause when newly emerged adults are exposed to short-day (12L:12D) photoperiods at 25 °C, whereas the adults reared under long-day (16L:8D) photoperiodsatthe same temperature begin reproduction within two weeks of exposure. Both sexes have almost identical responses to the short-day photoperiod [14]. The diapausingadultfemales of *R. pedestris* can be distinguished from the reproductive females by suppression in ovarian development and enlarged fat bodies inside of the abdominal cavity [15]. On the other hand, the reproductive system of the adult males has an enlarged ectodermal accessory gland reservoir, which is filled with secretory fluid under long-day conditions, but remains deflatedunder short-day conditions [16]. Previous studies found that the seminal vesicle was not present in this species, butsperm was found in the testes of both diapause and reproductive males, which is independent of mating status [17].

Although several physiological studies have beenconducted to investigatethe differences in the reproductive development of diapause and nondiapause females of *R. pedestris* adults, little has been described about the morphological development of internal reproductive organs in the diapause and reproductive insects of this species.Thus, prediapause and prereproductive *R. pedestris* adults were differentiated on the basis of the difference in the development of the accessory gland, ejaculatory duct, and size of the testis. Meanwhile, in the female adults, the length of ovarioles and thediameter of oviducts werealso measured.

## 2. Materials and Methods 

### 2.1. Insects Rearing and Sample Collection

More than a hundred pairs of *R. pedestris* were captured from the soya bean crop field of Huazhong Agricultural University (30°27′56″ N, 114°21′30″ E), Wuhan, China, during June 2017. This culture was further reared in 40 cm^3^ cages under controlled laboratory conditions at 25 ± 1 °C, R.H. of 70% ± 5%, a photoperiodic cycle of 16 h light (photophase) and 8 h dark periods (scotophase). Insects were fed with fresh soya bean pods (*Glycine maz*) and water (supplemented with 0.05% ascorbic acid solution), to maintain the population for many generations [13]. The newly laid eggs were collectedevery day and kept under the same conditions in an incubator (HP-250-GS, Wuhan Ruihua Instrument and Equipment, Wuhan, China). For hatching, the eggs were placed in 10 cm deep plastic boxes of 20 cm in diameter. Newly hatchednymphs were carefully collected with a camel brush and transferred into separate cages, and onlywater (as above) was supplied to the 1st instars of nymphs in 10 mL volumetric tubes. The tubes were plugged with cotton balls to facilitate the sucking of nymphs. 

### 2.2. Dissection and Imaging of Reproductive Organs

In the current study, *R. pedestris* induced diapause under ashort-day (12L:12D) photoperiod at 25 °C, whereas those exposed to along-day (16L:8D) photoperiod of the same temperature began reproduction after one week of adult eclosion. The abdominal cavitiesof 10 prediapause and prereproductive adults were dissected for each experimental group at the 1st, 3rd, 5th, 7th, and 9th day of adult emergence under a dissecting stereomicroscope (SMZ-t4, Chong Qing Optec Instrument, Chongqing, China) using 10× magnifications in phosphate-buffer saline (PBS, pH 7.4). Bugs were embedded into a dissecting petri dish (10 cm) using probe needles. Samples were washed three times with PBS solution and photographed with a mounted camera (Nikon D5100, Nikon Imaging Sales, Wuhan, China) over a stereomicroscope. 

### 2.3. Morphometric Analysis of Different Reproductive Parts

We classified the reproductive and diapause *R. pedestris* males based on the color of the accessory gland (AG). In our experiment, the males were considered as reproductive when the color of AG changed from whitish to yellowish-brown, whereas males having a whitish color of AG were classified as diapause males. On the other hand, the female reproductive system is well characterized, and it easy to distinguish between reproductive and diapause females on the basis of the developmental stage of the ovaries. The development of ovaries was classified into six stages as first stage ovaries have a hair-like structure of vitellarium with no development of blue yolk, while second stage ovaries have partially developed blue yolk and transparent hollow vitellarium and this is considered asdiapause. The remaining four stages are assumed to be reproductive (nondiapausing), and they are classified on the basis of numbers of blue oocytes developed in mature vitellarium [14]. The size of the accessory gland (mm^2^), the diameter of the ejaculatory duct (mm), length and width of the testis (mm)weremeasured in male adults, while in the female reproductive system, the length of vitellarium (mm) and the diameter of oviducts (mm) weremeasured using ScopePhoto 3.1 software (Appendix A).

### 2.4. Data Analysis

The significant differences between the reproductive parts of prediapause and prereproductiveadults of *R. pedestris* were calculated using the independent *t*-test with a 95% confidence level using Statistics Software SPSS 16.0 (SPSS Inc., Chicago, IL, USA). All the graphs were designed by GraphPad Prism 6.0 (GraphPad Software, Inc., San Diego, CA, USA).

## 3. Results 

### 3.1. Development of Internal Reproductive Organs in Prereproductive and Prediapause Females

The female internal reproductive system of *R. pedestris* is a very simple organ composed of two ovaries, a median oviduct, a common oviduct, and a spermathecal gland. The ovaries are ventrally located, and each is attached laterally withan elementary canal. There are nine tubular ovarioles in each ovary, which organizes a bunch-like structure posteriorly connected with each lateral oviduct. A couple of later oviducts are opened posteriorly into a common oviduct, and a pale-yellow colored spermathecal gland is dorsally located on the common oviduct. However, only one light-blue colored oocyte is developed inside each ovariole. Later on, these light-blue oocytes turn into the dark-brown color of mature eggs and are ovulated into the median oviduct of matured females. On the other hand, the ovarioles of immature females are without deposition of any blue oocytes. To investigate the effect of photoperiods, we compared the ovarian development in prediapause (12L:12D) and prereproductive (16L:8D) females after adult eclosion. The results showed that long-day(16L:8D)photoperiods promote reproductive development in prereproductive females, where their ovaries were developedsystemically and filled with blue yolk deposition (Figure 1A), whereas the ovarian development in prediapause females wassuppressed under short-day (12L:12D) photoperiods throughout the nine days of observation (Figure 1B). 

#### 3.1.1. Length of Vitellarium

Each ovariole of *R. pedestris* females is composed of two parts: the distal germarium produces the primary oocytes by a mitosis program, and the proximal tubular part (namedvitellarium) is responsible for housing and maturing the primary oocytes by a vitellogenesis process [18]. The length of a vitellarium has a direct relationship with the development of the yolk. The shape of vitellarium frequently changes from hair-like structures to hollow transparent rectangular sacs in prereproductive *R. pedestris* females. Later on, this rectangular sac is changed to an oval sac, with thestorage of blue yolk, under constant long-day (LD) photoperiod conditions, whereas, prediapause females exhibit considerable pressure on reproductive development and maintaina hair-like lobe of vitellarium throughout short-day (SD) photoperiodic exposure(Figure 1C). At the very beginning, from 3 days after adult eclosion, prereproductive females attained a significantly larger size of vitellarium than prediapause females (t = 3.081, df = 18, *p* = 0.017; Figure 2A).

#### 3.1.2. Diameter of the Lateral Oviduct 

The female reproductive system containsa pair of lateral oviducts arising from the posterior side of the ovarioles in which mature eggs arestored. These lateral oviductsare used for transporting the eggs ovulated in thevitellarium to the ovipositor. Towards maturity, the muscles of oviducts are more expanded and provide the space for egg storage. The diameter of each lateral oviduct is frequently expanded with the age of prereproductive females, whereas in the prediapausing females, it remained immaturethroughout 9 days of SD photoperiodic exposure (Figure 1C). The difference between the diameters of both prediapause and prereproductive females of *R. pedestris* showed significant difference after 5 days of adult eclosion (t = 4.673, df = 18, *p* < 0.01; Figure 2B).

### 3.2. Development of Internal Reproductive Organs in Prereproductive and Prediapause Males

The male reproductive system of *R. pedestris* adults consists of a pair of testes, vas deferens, accessory glands, and ejaculatory duct. Two oval-shaped testes are suspended on the ventral side of the elementary canal. The outer surface of the testes is covered with a reddish color peritoneal sheath. Each of thetestis lobesisposteriorly attached with a vas deferens, which is a uniform tubular structure connected toan ejaculatory duct. A very prominent transparent sac dorsally attached to the ejaculatory duct is known as AG. The color of AG continuouslychanges from whitish to yellowish-brown in reproductive males. It was observed that the long-day photoperiod promoted reproductive development in prereproductive males of *R. pedestris* throughout observations, whereas the SD photoperiod ceased the reproductive development in prediapause males, and the accessory gland remained immature (Figure 3B).

#### 3.2.1. Size and Color of Accessory Gland

In males, the accessory gland is the sac-like structure having the function of producing and nourishing unreleased sperm. This sac-like structure becomes larger during the spermatogenesis process but remains immatureunder stress conditions. Different photoperiodic conditions significantly influenced the development of this part. The size of the accessory gland was increased, and the color was changed whitish to yellowish-brown when males were kept under LD conditions, whereasthe SD conditions suppressed the growth of AG and maintained its smaller size (Figure 3C). However, a clear significant difference was observed between the size of the accessory gland of prereproductive and prediapause males after 5 days of adult emergence (t = 3.495, df = 18, *p* = 0.014; Figure 4A).

#### 3.2.2. Diameter of Ejaculatory Duct

The ejaculatory duct is a cylindrical tube that is connected withAG by its posterior head, and the other head is open as external genitalia. The diameter of the ejaculatory duct increased up to a specific level and then maintained its growth in a straight line under long-day photoperiodic conditions, whereas short-day conditions not only inhibited the development of this duct but also shrank the diameter (t = 2.324, df = 18, *p* = 0.032; Figure 4B).

#### 3.2.3. Length and Width of Testes 

The male reproductive system of *R. pedestris* has two oval shape lobes hanging by each vas deferens, known as testes. The main function of the testes is the storage and the production of sperm cells, along with several hormones, in its sac. It was observed that the length and width of the testes were developed differently and irregularly under both photoperiodic conditions (SD and LD). It is interesting to mention here that the size of the testes was bigger during the early days of both conditions, but the developmental differences between prereproductive and prediapausemale testes werenot significant throughout the experiment (t = 1.228, df = 18, *p* > 0.05 and t = 0.790, df = 18, *p* > 0.05 length and width, respectively; Figure 4C,D).

## 4. Discussion

The internal reproductive system of female *R. pedestris* is very simple and identical to *Perillus bioculatus*—a pair of ovaries containing nineovarioles in each ovary, which arefused in each median oviduct. Both median oviducts posteriorly combine to form a common oviduct on which a spermathecal gland is attached. Previous studies have shown that there are sevenovarioles in each ovary of *P. bioculatus*, which would be different in numbers in the different species of true bugs [19]. The females of stink bugs (*Bagrada hilaris*) contain 5–7 meroistic and telotrophicovarioles since specialized nursing cells are present within the germarium, which provides the nutrients to each follicle through nutritive loops [18].The ovarioles in *R. pedestris* females are composed of two regions, i.e.,germarium and vitellarium glands. So, it can be supposed that the females of *R. pedestris* also contain conserved telotrophicovarioles that nourishing the follicles through thegermarium and house the oocytes in the vitellarium. Unlike telotrophicovarioles, the polytrophic ovariole is a continuous process and produces many oocytes. Moreover, polytrophic ovarioles do not have specialized nursing cells, which is very common in primitive insects [18]. In general, the ovaries of the reproductive insects develop in a synchronized manner and mature at a particular stage when environmentalconditions become favorable for their survival [20]. However, the ovaries of *R. pedestris* females have a systemic manner of development, which can be separated into two distinct categories: parous (reproductive) and nulliparous (nonreproductive), depending on the presence or absence of oocytes in the vitellarium of ovarioles, respectively. It has been found that the prediapause *R. pedestris* females undergo suppression periods of ovarian development and acquire nulliparous ovaries. The ovarioles in prediapausefemales were undifferentiated and appeared as long, transparent tubes; empty and smaller sizedvitellarium was present throughout the investigations. In comparison, the ovarioles in prereproductive females were continuously developed, and blue color oocytes could be differentiated in parous ovaries. These differences in the development of ovaries were based oncharacterizingprediapause and prereproductive females.

In contrast, the male reproductive systems are very similar in diapause and reproductive insects and hard to differentiate without analyzing at a micro-level [21]. Some researchers have categorized the diapausing and nondiapausing males based on the size and color of reproductive organs [22]. The color and the shape of AG frequently changes along the development of *R. pedestris* males. At the beginning of reproductive development, a very small whitish milky AG changes to a well-expanded yellowish-brown AG in coloration. In our investigation, we found that the prediapause males limited their reproductive development and developed smaller sized reproductive parts, whereas the prereproductive males constantly developedtheir reproductive organs under continuous exposure of LD. In a different study on *D. melanogaster*, the males were kept under dormant conditions for 3 weeks and the morphological development of the reproductive organs was studied after the dormant period. In dormant males, the size of accessory glands and seminal vesicles weresmaller compared to the males that were reared in nondiapausing conditions. It was found that spermatogenesis is arrested in dormant males and sperm cells in seminal vesiclesarevery few compared to reproductive males [23,24]. Although the seminal vesicle is absent in the male reproductive system of *R. pedestris*, sperm was present in the reproductive and diapausing males [16,17]. Similarly, in the predatory stinkbug *Podisus nigrispinus*, the seminal vesicle is also absent in the reproductive system ofmales.Spermatozoa were noticed in the lumen of vas deferens, which is also independent of mating status. Moreover, all the phases of spermatogenesis were observed in the testicular follicles of newly eclosed *P. nigrispinus* adults and just after the mating process. It has been found that the spermatozoa were continuously produced even during the mating state, and the newly eclosed males store the spermatozoa in the lumen of the vas deferens [25]. In the pupal stage of stingless bees (Apidae: Meliponini), the male accessory glands are absent and the secretory function of this organ is performed by their seminal vesicle, whereas in the adult males, the accessory glands also store the spermatozoa [26].In the current study, the development of the accessory gland and ejaculatory duct in prediapause males was poor compare to prereproductivemales. It can be argued that the delay in reproductive development may result in the absence of hormonal productionduring the diapause period, which needs to be explored in future investigations.

## 5. Conclusions 

In conclusion, we found that the prediapause and prereproductiveadultsof *R. pedestris* can be categorized on the basis of the development of the internal reproductive organs because the diapause bugs arrest reproductive organs and develop as newly emerged adults of reproductive insects. Hence, the above findings not only provide basic knowledge to distinguish diapause and reproductive *R. pedestris* adults but are also applicable in further molecular investigations.

## Figures and Tables

**Figure 1 insects-11-00347-f001:**
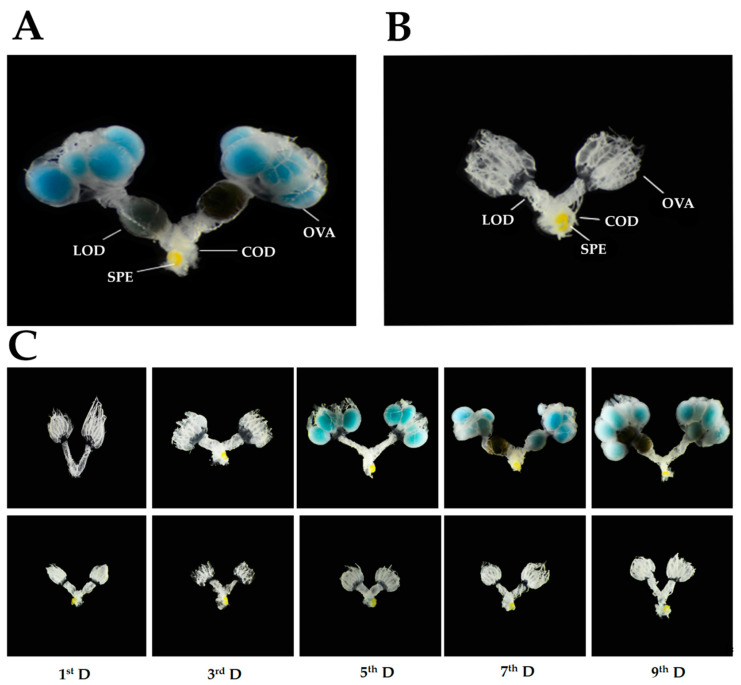
Development of the internal reproductive organs of the prereproductive (**A**) and prediapause females (**B**) after 9 days of adult emergence. Differences in the development of ovaries in the prereproductive (upper) and prediapause (lower) adult females of *R. pedestris* at each day of observation (**C**). Ovariole(OVA), lateraloviduct(LOD), common oviduct(COD), spermatheca(SPE).

**Figure 2 insects-11-00347-f002:**
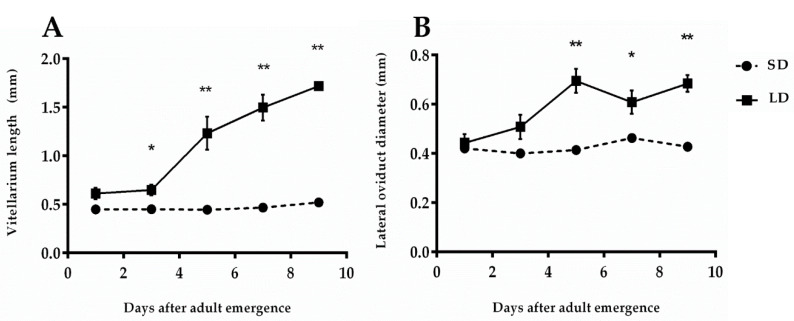
Differences in the development of different parts of ovaries in prediapause (LD) and preoviposition (SD) females of *R. pedestris*. The horizontal lines indicate the experimental days after adult emergence, whereas perpendicular lines show the measurements of vitellarium length (**A**) and lateral oviduct diameter (**B**). Each point represents the mean ± SEM for 10 individuals. Asterisks indicate significant differences between the independent samples according to *t*-test,* *p* < 0.05, ** *p* < 0.01.

**Figure 3 insects-11-00347-f003:**
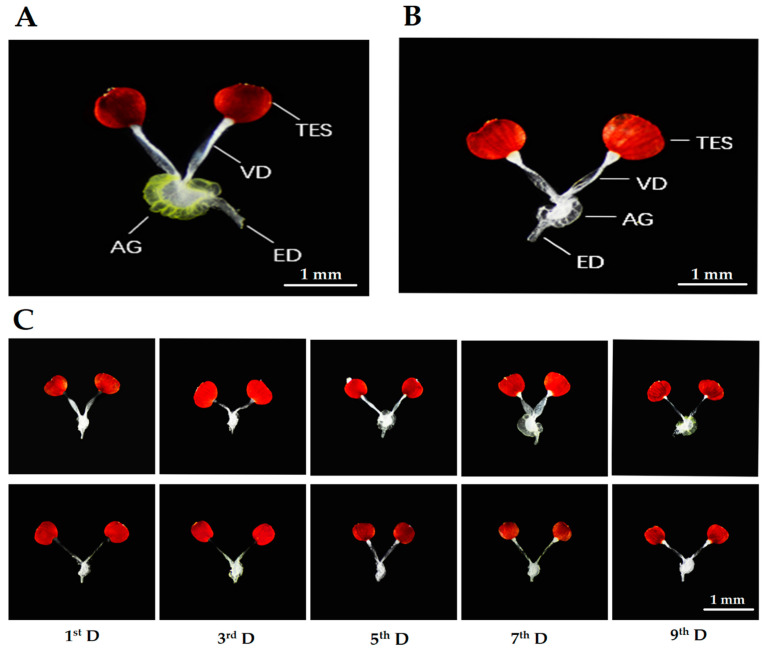
Development of internal reproductive organs of prereproductive (**A**) and prediapause (**B**) males after 9 days of adult emergence. Differences in the development of prereproductive (upper) and prediapause (lower) of *R. pedestris* male adultsat each day of observation (**C**). Accessory gland (AG), vas deferens (VD), testis (TES), ejaculatory duct (ED).

**Figure 4 insects-11-00347-f004:**
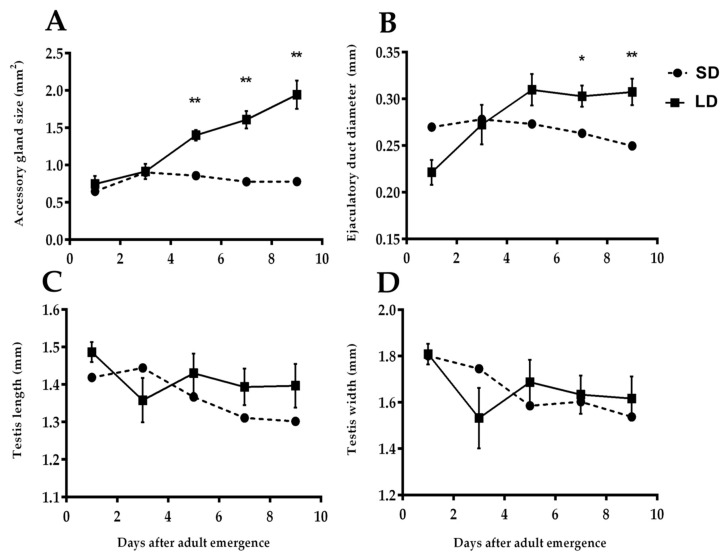
Difference in the development of different reproductive parts in prediapause (**SD**) and prereproductive (**LD**) males of *R. pedestris*. The parallel lines of graphs indicate the experimental days after adult emergence, whereas perpendicular lines show the development of accessory gland (**A**), ejaculatory duct (**B**), and testes of length (**C**) and width (**D**). Each point represents the mean ± SEM for 10 individuals. Asterisks indicate the significant differences between independent samples according to *t*-tests, * *p* < 0.05, ** *p* < 0.01.

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
