# Peer review of "Developmental Differences on the Internal Reproductive Systems between the Prediapause and Prereproductive Riptortus pedestris Adults"

_insects, 2020, doi:10.3390/insects11060347_

Round 1
Reviewer 1 Report
The authors examined about the internal morphology of reproductive organs of both sexes to compare between pre-diapause and pre-reproductive adults in Riptortus pedestri. This bug, and closely related species Riptortus clavatus have been well investigated by Numata. In his papers, I found description on the morphology of male reproductive organs of this bug (Ikeno T., Numata, H., & Goto, S.G. (2011) Circadian clock genes period and cycle regulate photoperiodic diapause in the bean bug Riptortus pedestris males. Journal of Insect Physiology 57: 935-938. Moreover, Spermatogenesis did not change between diapausing and reproductive adult males (Numata, H. & Kobayashi S. (1989) Morphological and behavioral character of adult diapause and its termination by a juvenile hormone analogue in Riptortus clavatus. Tonner, M.,/Soldan,T./Bennettova,B.(eds.) Regulation of insect reproduction ⅣProceedings of a symposium held in Zinkovy, September 1987, Academia Praha 1989: 401-411. Although the latter paper may be difficult to obtain, it is better that the former paper is quoted. Of course, no quotation of these papers will not lower the value of this manuscript. However, I would like to know about the basic life history of this bug. When do they mate? After overwintering only? Are they polyvoltine? When does spermiogenesis start? Final nymphal stage or adult stage or both? Why do diapausing adult males have mature sperm before overwintering? I feel this might raise the possibility that they may copulate before overwintering depending on the environmental conditions.
In addition, I recommend the authors to mention testis produce and store the sperm in Introduction or Materials and Methods. I mean they do not have a seminal vesicle.
Author Response
We are very thankful to reviewer 1 for the helpful suggestions. We have taken all these comments and suggestions into account and revised the manuscript

Reviewer 2 Report
Comments in file attached

Author Response
Comments and Suggestions for Authors from Reviewer 2
Review of “Developmental differences on the internal reproductive systems between the pre-diapause and pre-reproductive Riptortus pedestris adults”
By Hafeez et al.
General comments
The article is overall well written although I feel the use of English could be improved in some aspects. The introduction is a bit short and it would improve the paper if more information is given on the importance of studying reproductive diapause in insects, both in an ecological context and an evolutionary and developmental biology context. Results are clear and well described.
I cannot make very specific comments because I do not have access to line numbering in the article.
Introduction
In the first paragraph of introduction, I would like to see one more sentence with information on the adverse conditions generally initiating reproductive diapause in Insects (temperature, photoperiod). In some species it can be the lack of food, or drought.
In the second paragraph, it is not clear when diapause is expressed because it is mentioned that both the adults and the nymph can enter diapause. Is the species able to express both immature diapause and adult reproductive diapause? In this case, what determines the development stage at which it is induced? Also, this paragraph is more focused on the biological material and not on the subject. It is fine unless the introduction is unbalanced, as it stands now, because information on the main topic of the paper (reproductive diapause) is missing.
The last paragraph of the introduction lacks information on what is already known on internal morphological differences in diapausing insects. It has already been investigated in several species and should be mentioned and detailed here. What is usually observed after a few days of reproductive diapause compared to maintenance of reproduction? How the vitellarium length and the oviduct diameter usually change across the days? Do the authors expect differences with what is already known on other insects?
Methods
It would be interesting to also analyse how changes in reproductive parts vary among time? Authors show significant differences between diapause and non-diapause, but do traits showed in Figure 2 significantly change over time and starting which day is it significant? It could help future studies determining when a dissection should be conducted to reveal the presence of diapause.
Discussion
No need to start with “apparently”.
Some of the information provided in the beginning of the discussion could be placed at the end of the introduction.
We are very thankful to reviewer 2 for the valuable suggestions. We have taken all these comments and suggestions into account and revised the manuscript according to his suggestions:
- Comments and Suggestions for Authors
The article is overall well written although I feel the use of English could be improved in some aspects.
Response
Yes, we have put special concern about the English now.
- Comments and Suggestions for Authors
In the first paragraph of introduction, I would like to see one more sentence with information on the adverse conditions generally initiating reproductive diapause in Insects (temperature, photoperiod). In some species it can be the lack of food, or drought.
Response
We have added detailed information about the reproductive diapause and the factors which responsible in induction of diapauseand the strategies to overcome on the adverse affect of environmental conditions.First paragraph, Line no is 2-8 with reference of 2-4
Moreover, we added some sentences about the differentiation of diapause phase in the first paragraph of introduction. First paragraph, Line no is 13-19 cited with reference of 7-9
- Comments and Suggestions for Authors
In the second paragraph, it is not clear when diapause is expressed because it is mentioned that both the adults and the nymph can enter diapause. Is the species able to express both immature diapause and adult reproductive diapause? In this case, what determines the development stage at which it is induced? Also, this paragraph is more focused on the biological material and not on the subject. It is fine unless the introduction is unbalanced, as it stands now, because information on the main topic of the paper (reproductive diapause) is missing.
Response
In the second paragraph, we changed the wordings which created the confusion and now it clear about “The physiological studies showed that both nymph and adults of the R. pedestris are sensitive to environmental factors including food, temperature and photoperiod. However, the investigations found that the fourth and fifth instar nymphs are more sensitive than adult to the photoperiods. The R. pedestris insects entered the reproductive diapause when newly emerged adults were continuously exposed to short-day (12L:12D) photoperiod at 25°C, whereas the adults reared under long-day (16L:8D) photoperiod of the same temperature began reproduction within two weeks of exposure”. Starts from the line no. 6-10 and cited with 12-13 reference
Moreover, we added some sentences about the male reproductive diapause in the second paragraph Starts from the line no. 12-16 and cited with 15 reference
- Comments and Suggestions for Authors
The last paragraph of the introduction lacks information on what is already known on internal morphological differences in diapausing insects. It has already been investigated in several species and should be mentioned and detailed here. What is usually observed after a few days of reproductive diapause compared to maintenance of reproduction?
ResponseWe added the detail information about what have done before in this species in paragraph no 2 in introduction with line no 66-78 cited with reference no. 12-17
Comments and Suggestions for Authors
How the vitellarium length and the oviduct diameter usually change across the days? Do the authors expect differences with what is already known on other insects?
Response
Yes the vitellarium length and the oviduct diameter usually change across the days because there was some variation in the size of these structures largely due to the signals received during the initiation of diapause phase and preparation of reproductive status. In addition, the size of the vitellarium is dependent on the maturity of the most proximal follicle which increases in length as it matures.
- Comments and Suggestions for Authors
Methods
It would be interesting to also analyse how changes in reproductive parts vary among time? Authors show significant differences between diapause and non-diapause, but do traits showed in Figure 2 significantly change over time and starting which day is it significant? It could help future studies determining when a dissection should be conducted to reveal the presence of diapause.
Response
Yeah all the parts are showing variation by the observation time which mainly depends on the token stimuli during the induction phase of diapause because all the individuals respond and prepare the internal development according to what they percept during nymphal and the observation days. But as per observation, 5th day is the most commonly significant among all the reproductive parts of male pre-diapause and pre-reprodutive male and females of this species.
- Comments and Suggestions for Authors
Discussion
No need to start with “apparently”.
Response
We have deleted it from the start of discussion.
- Comments and Suggestions for Authors
Some of the information provided in the beginning of the discussion could be placed at the end of the introduction.
Response
Yes, we have lengthen the introduction with all the paragraphs

Reviewer 3 Report
The manuscript entitled „Developmental differences on the internal reproductive systems between the pre-diapause and pre-reproductive Riptortus pedestris adults“ brings results of a simple description of gonads in diapause and non-diapause adults of Riptortus pedestris.
The main problem of the manuscript is overall low scientific soundness. A diapause is actually the reproductive diapause, defined by the arrestment of gonads development. It is totally not surprising that diapause adults have less developed gonads than the non-diapause reproductive individuals. There are plenty of papers where diapause vs. reproductive individuals are distinguished on the basis of gonads development (see e.g. Spurgeon, D. W., Sappington, T. W., & Suh, C. C. (2003). A system for characterizing reproductive and diapause morphology in the boll weevil (Coleoptera: Curculionidae). Annals of the Entomological Society of America, 96(1), 1-11; Ditrich, T., & Kostal, V. (2011). Comparative analysis of overwintering physiology in nine species of semi‐aquatic bugs (Heteroptera: Gerromorpha). Physiological entomology, 36(3), 261-270) – this information should be at least mentioned in Introduction.
Beside low scientific soundness, the manuscript is relatively clear and well structured. The Introduction seems to contain also some results (last sentence of the penultimate paragraph – the lack of line numbering makes the review somewhat confused).
The Results section contains well-presented results of the study. I am not sure if really all t-tests were significant with P EQUAL to 0.05 (maybe the authors wanted to write P < 0.05? Or were really all Ps – 0.05? Check the results and provide the exact value of p (rounded to e.g. 2 decimal places). Just a minor remark – the “later” on the 5th line of the first paragraph of the Results should probably be “lateral”.
The Discussion and Conclusion should contain some information on testes development – what do the results show about development of male gonads, if can be diapause and reproductive males distinguished upon the state of their gonads. The very brief conclusion is actually present in the Introduction.
The text needs extensive English editing; I have not corrected English in this review.
Author Response
Comments and Suggestions for Authors
The manuscript entitled „Developmental differences on the internal reproductive systems between the pre-diapause and pre-reproductive Riptortus pedestris adults“ brings results of a simple description of gonads in diapause and non-diapause adults of Riptortus pedestris.
The main problem of the manuscript is overall low scientific soundness. A diapause is actually the reproductive diapause, defined by the arrestment of gonads development. It is totally not surprising that diapause adults have less developed gonads than the non-diapause reproductive individuals. There are plenty of papers where diapause vs. reproductive individuals are distinguished on the basis of gonads development (see e.g. Spurgeon, D. W., Sappington, T. W., & Suh, C. C. (2003). A system for characterizing reproductive and diapause morphology in the boll weevil (Coleoptera: Curculionidae). Annals of the Entomological Society of America, 96(1), 1-11; Ditrich, T., & Kostal, V. (2011). Comparative analysis of overwintering physiology in nine species of semi‐aquatic bugs (Heteroptera: Gerromorpha). Physiological entomology, 36(3), 261-270) – this information should be at least mentioned in Introduction.
Beside low scientific soundness, the manuscript is relatively clear and well structured. The Introduction seems to contain also some results (last sentence of the penultimate paragraph – the lack of line numbering makes the review somewhat confused).
The Results section contains well-presented results of the study. I am not sure if really all t-tests were significant with P EQUAL to 0.05 (maybe the authors wanted to write P < 0.05? Or were really all Ps – 0.05? Check the results and provide the exact value of p (rounded to e.g. 2 decimal places). Just a minor remark – the “later” on the 5th line of the first paragraph of the Results should probably be “lateral”.
The Discussion and Conclusion should contain some information on testes development – what do the results show about development of male gonads, if can be diapause and reproductive males distinguished upon the state of their gonads. The very brief conclusion is actually present in the Introduction.
The text needs extensive English editing; I have not corrected English in this review.
- Comments and Suggestions for Authors
The main problem of the manuscript is overall low scientific soundness. A diapause is actually the reproductive diapause, defined by the arrestment of gonads development. It is totally not surprising that diapause adults have less developed gonads than the non-diapause reproductive individuals. There are plenty of papers where diapause vs. reproductive individuals are distinguished on the basis of gonads development (see e.g. Spurgeon, D. W., Sappington, T. W., & Suh, C. C. (2003). A system for characterizing reproductive and diapause morphology in the boll weevil (Coleoptera: Curculionidae). Annals of the Entomological Society of America, 96(1), 1-11; Ditrich, T., & Kostal, V. (2011). Comparative analysis of overwintering physiology in nine species of semi‐aquatic bugs (Heteroptera: Gerromorpha). Physiological entomology, 36(3), 261-270) – this information should be at least mentioned in Introduction.
Response
As per reviewer’ advice and suggestions, we added detail information in introduction line no 2-8 cited with 1-3 reference
Wealso added detail information in introduction on differentiation of diapause phase and the factors which responsible for induction of dipause line no 14-20 cited with 7-9 reference
Moreover we also added some information on male reproductive parts in paragraph no. 2 with line no. 12-15 cited with reference no. 15
- Comments and Suggestions for Authors
The Results section contains well-presented results of the study. I am not sure if really all t-tests were significant with P EQUAL to 0.05 (maybe the authors wanted to write P < 0.05? Or were really all Ps – 0.05? Check the results and provide the exact value of p (rounded to e.g. 2 decimal places). Just a minor remark – the “later” on the 5th line of the first paragraph of the Results should probably be “lateral”.
Response
Yes we wanted to write the P < 0.05 and have different p-values for each reproductive organ, but this was overlooked before and has nowmentioned the exact values as
The p = 0.017 value for the length of vitellarium, for the Diameter of the lateral oviduct is p = 0.00, the Size of accessory gland has p = 0.14, the Diameter of Ejaculatory duct is significantly different at p = 0.032, whereas the length and width of testes are not significantly different and both has p > 0.05 value.
- Comments and Suggestions for Authors
Just a minor remark – the “later” on the 5th line of the first paragraph of the Results should probably be “lateral”.
Response
Yes your observation was right and on the 5th line of first paragraph should be Lateral oviduct not later oviduct.
- Comments and Suggestions for Authors
The Discussion and Conclusion should contain some information on testes development – what do the results show about development of male gonads, if can be diapause and reproductive males distinguished upon the state of their gonads. The very brief conclusion is actually present in the Introduction
Response
We acknowledged the suggestions of reviewer and added the relevant information about spermatogenesis mechanism of true bugs in second paragraph from line no 15-26 and cited with 22-25 references. Moreover we put some changes in conclusion to make it clear for the common readers.

Reviewer 4 Report
It is an interesting topic. The paper will be useful for diapause studies of true bugs. The work is well illustrated.
However, I cannot advise to accept the MS as it is because the text needs serious editorial work. There are a few mistakes even in the spelling of the model species, in legends of figures, in References, etc. English needs to be carefully checked.
I added a lot of comments and corrections in the pdf file.

Author Response
We are very thankful to reviewer 4for the helpful suggestions. We have taken all these comments and suggestions into account and revised the manuscript as follows:
Comments and Suggestions for Authors from Reviewer 4 from Abstract
- Alydidae with link
- Add facultative
Response
- As per reviewer’ advice, these was no need to add link with Alydidae here, so we removed the link from there
- We added the facultative before reproductive diapause
Comments and Suggestions for Authors from First paragraph
- Add facultative word between Undergo diapausing state
- How is it unique if it is in all orders?
- Why several? Do you know any exception?
- These are not three different orders. It is either Hemiptera, OR Heteroptera + Homoptera
- Resume what?
- But in some species diapause is not facultative, it is obligatory....
- Nature of= omit
- During diapause induction phase, several insects undergo a plastic response known as facultative diapause in which diapause initiation is determined by the receipt of specific environmental signals.
Response
- Rewords the sentence = Under adverse conditions of the environment, insects undergo an alternative period of suppression in the reproductive development rather developing into reproductive one.
- We didn’t found any species of insect which do not have diapause. We change word unique to common.
- We have changed the several to all.
- We have rewritten the sentence and write the detail information about the reproductive diapause in insects and the factor causing the reproductive diapause in insect species.
- We have removed the word nature from the sentence.
- As per suggestion of reviewer, we used the order name Hemiptera instead of three names separately.
- We have removed the word resume from the sentence and rewritten sentence
- We rewrote sentence and added detail information about initiation of diapause by environmental factors in line no. 14-17, cited with reference no. 8
Comments and Suggestions for Authors from Second paragraph
- Riptortus pedestrian
- Facultative reproductive diapause
- Those were reared= who reared ?
- (12L: 12D)= remove the spacing
- Begin or began?
- Pedestris = lowercase
- Large number of fat bodies= enlarge, developed or large
- Sperms were found or sperm was
- Both diapause and non-diapause males = add reference
- Examination was limited to female= where?
- Between the pre-diapause and pre-reproductive males= reference ?
Response
- The name of species name was overlooked and now changes the spelling of species name from Riptortus pedestrian to Riptortuspedestris.
- As per suggestion of reviewer, we used added the word facultative in reproductive diapause in paragraph no 2 of introduction and line no. 2
- Line no 8 of Paragraph no. 2 in introduction, information was missing about the rearing the insects. Now added the information about the insects and their conditions under they were reared.
- There was spaces between all the conditions written in brackets as (12L: 12D) (16L: 8D), now we removed the spaces as (12L:12D) (16L:8D).
- The words Begin was mistakenly used and now we change the first form to second form and used began in change of begin.
- the species name was overlooked and wrote in uppercase Pedestris, now we use the species name with lowercase as R. pedestris.
- The word number has changed and now used enlarge.
- The comment and suggestion number 16-19 came under one sentence and all the comments and sugessions fulfill in one sentence cited reference no. 15 at the very end of paragraph no. 2 in introduction.
Comments and Suggestions for Authors from Third paragraph
- Although several physiological studies have been investigated to differentiate the diapause and non-diapause females of this species, there is still unknown how internal morphological development of reproductive organs is being developed by diapause and reproductive insects = rewords
- Testis was measured in= English check
- While, in the female= English check
Response
- The comment and suggestion number 20-22 came under one paragraph and suggestions were about too reword the sentences, so have rewords the last paragraph of introduction.
Comments and Suggestions for Authors from Reviewer 4 from Materials and Methods
Comments and Suggestions on First paragraph
- This heterogeneous population = meaning ?
- 25±1oC, R.H of 70±5% = Add dot after R.H
- And water (0.05% ascorbic acid solution) = Add word, supplement with
- Water (0.05% ascorbic acid solution) supplied = don’t use again
Response to Comments and Suggestions
- We have used the word “ this culture instead of “the heterogeneous population”
- We used the dot after R.H abbreviation of relative humidity
- As per reviewer’ advice, we added the word “supplement with” between water (0.05% ascorbic acid solution)
- As per reviewer’ advice, we removed the detail information on water and used “as above” instead of (0.05% ascorbic acid solution)
Comments and Suggestions on second paragraph
- In current = “the” should use
- pedestrian = check the species name
- The SD photoperiod = specify the hours of light and dark
- To LD photoperiod = specify the hours of light and dark
- week of adult eclosion = use after not of
- (PBS, pH7.4) = use space
- Nikon Imaging (China) Sales, Wuhan, China) over stereomicroscope = check it
Response to
- We added “the” between In current
- We corrected the species name as pedestrisinstead of R. pedestrian
- We wrote the detail information abou the SD photoperiod and specify the light and dark hours for SD photoperiod
- We wrote the detail information abou the SD photoperiod and specify the light and dark hours for LD photoperiod
- We used “after” instead “of”
- We added space between (PBS, pH7.4)
- Before it was used “China” two times in “Nikon Imaging (China) Sales, Wuhan, China) over stereomicroscope” and now has written as “Nikon Imaging Sales, Wuhan, China) over stereomicroscope”
Comments and Suggestions on third paragraph
- male basis on the = based on the
- color of AG was changed from = correct the English
- of development of ovarian stage = development of avaries or developmental stage of ovaries
- development was classified into six stages = check the English
- being in reproduction = being in diapause
- testis was measured = use “were”
- oviduct (mm) was measured= use “were”
Response to
- we corrected the grammar as “based on the” instead of “male basis on”
- we corrected the grammar as “AG changed” instead of “AG was changed”
- we used the sentence as of “of developmental stage of ovaries” instead of “development of ovarian stage”
- we corrected the English
- we corrected right word as “being in diapause” instead of “being in reproduction”
- we corrected the grammar mistake as “were “ instead of “was”
- we corrected the grammar mistake as “were “ instead of “was”
Comments and Suggestions on fourth paragraph
- pre-reproductive insects of pedestris = use “adults”
- student t-test with = use “the” before
Response to
- we used “Adults” instead of “insects”
- we used correct name of test as “the independent t-test” instead of “student t-test“
Comments and Suggestions for Authors from Reviewer 4 from Results
Comments and Suggestions on First paragraph
- Of pedestrians = correct the species name?
- bunch like = use hyphen
- Color of oocyte = ?
- Were infertile or without = not fertilized? I do not think you can say so...
- (SD) and pre-reproductive (LD) females = justify
- deposition after each observation
- Figure 1 = Photos of Day 9 are basically the same as A and B --- What is the reason to show both pairs then?
- pedesretisafter each = at
- observation (C). Ovary (O): Ovariole (OVA): Oocyte (OC): Lateral Oviduct (LOD): Common Oviduct (COD) and Spermatheca (SPE).
Response
- We corrected the species name as pedestris
- We used hyphen between bunch like
- We removed “of”
- We removed the “infertile” and now it is clear to reader
- We wrote the detail hours of lond-day and short-day
- We removed the “after each observation”
- Figure 1 are different in “picture A is representing the development of pre-reproductive females, whereas picture A is representing the development of pre-diapause
- We used “at” instead of “after”
- Pictures were labeled with other abbreviations and now change the abbreviations in pictures
Comments and Suggestions on second paragraph
- hair like= hyphen
- At very = the
Response
- We use hyphen to make it correct as hair-like
- We used “the” between “at very”
Comments and Suggestions on third paragraph
- from 10 independent biological replicates of each observation = what does it mena? 10 adults? if so, say so directly
- samples t-test = according to the
Response
- Now we used simple words to explain as 10 adults
- As per reviewer advice we used according to
Comments and Suggestions on fourth paragraph
- bug ( pedestris), is a very = font
- Fine structure = what do you mean?
- Each of testes lobe = font
- brown depends on the = check English
- Figure 3. internal = uppercase
- Of reproductive (upper) and diapause = pr pre-reproductive? And pre-diapause
Response
- We corrected the font size
- We used simple structure instead of fine structure
- We changed the font size
- We rewrote the sentence
- Change the lowercase to uppercase
- Before it was overlooked and now has change it as pre-reproductive and pre-diapasue
Comments and Suggestions on fifth paragraph
- the size and turns = check English
- clear has been observed = check English
Response
- We have structured the sentence
Comments and Suggestions on fifth paragraph
- duct (ED) is a cylindrical = you did not give abbreviations in previous sections
- This is the most important part in male reproductive system =It is a pretty strange statement... Do you think the system could work without any other section? No. aAll parts are important here.
- Diameter of ED was = ?
- but also shrunken the = bad word
Response
- We checked it and used and abbreviation in previous sentences
- We removed the bad words which over expressed the importance
- We used full name with abbreviation
- We removed the bad word from sentence
Comments and Suggestions on sixth paragraph
- system of pedestrians has = species name
- Between pre-reproductive and pre-reproductive = ?
- Was not much significant throughout = ?
- graphical mistake = G capital
- graphical mistake = D capital
- 10 independent biological replicates =?
Response
- We used correct species name as “ pedestris” instead of R. pedestrians
- It has been overlooked and now change it with pre-reproductive and pre-diapause
- We removed much from the sentence
- We change d the graphical mistake as no uppercase G
- We change d the graphical mistake as no uppercase D
Comments and Suggestions for Authors from Reviewer 4 from Discussion
Comments and Suggestions on first paragraph
- And once produced many oocytes.
Response
We have restructured the sentence
Comments and Suggestions on second paragraph
- Diapausing males based = of other species! Threre are papers on diapause in males of heteropterans, e.g. Nezaraviridula with a lot of details.
Response
As per reviewer’ suggestion we added more details on some closely related species and the recommended species
Comments and Suggestions for Authors from Reviewer 4 from Reference
Response
We accepted all the suggestion and changed all the mistakes from the references

Round 2
Reviewer 3 Report
The revised version of the manuscript is improved in comparison to the first version. If editors consider the topic as appropriate for publication in Insects , I think the manuscript can be accepted in the present form (after slight English editing).
Author Response
We are very thankful to reviewer 3 for considering our manuscript for acceptance. We have taken all these comments and suggestions into account and revised the manuscript as follows:
Comments and Suggestions for Authors from Reviewer 3
The revised version of the manuscript is improved in comparison to the first version. If editors consider the topic as appropriate for publication in Insects, I think the manuscript can be accepted in the present form (after slight English editing).
Response
Yes we checked and improved English weakness by English checker agency as mywritinghelp
mressayhelp@yahoo.com with reference number: '2020-D-225

Reviewer 4 Report
Many points are corrected, but it seems that English became worse. Authors MUST find somebody who knows biology and ask him/her to seriously check and improve English. Also, authors discuss ecophysiology (especially, in the Introduction), but it seems that they do not understand it well. Adult (=reproductive) diapause is only one type of diapause. It shold be kept in mind that the proceses involved in embryonic, larval/nymphal, pula diapauses are different.
All other comments are in the attached pdf file.

Author Response
We are very thankful to reviewer 4for the helpful suggestions. We have taken all these comments and suggestions into account and revised the manuscript as follows:
Many points are corrected, but it seems that English became worse. Authors MUST find somebody who knows biology and ask him/her to seriously check and improve English. Also, authors discuss ecophysiology (especially, in the Introduction), but it seems that they do not understand it well. Adult (=reproductive) diapause is only one type of diapause. It shold be kept in mind that the proceses involved in embryonic, larval/nymphal, pula diapauses are different.
Response
Yes we checked and improved English mistakes by mywritinghelp
While improving this manuscript this time, we putt special concern on adult diapuse
Comments and Suggestions for Authors from Reviewer 4 from Abstract
into facultative diapause under = facultative adult (or reproductive) diapause
there is little known about = little is known
vitellariums = or vitellaria? check this
whereas = why? I woud say ''and''
Response
- We changed it as reproductive adult diapause
- we corrected it as little is known
- We mentioned here it for singular as vitellarium
- We corrected the grammar mistake and used “and”
Comments and Suggestions for Authors from First paragraph
- However = Why However?
- or photoperiod, failure of mating, absence = strictly speaking, short day-length is not an adverse condition, it is a signal. Also failure of maiting is not adverse condition as well...
- the adversity from the danger, insects = strage wording
- adopted the several = not THE -- English needs to be checked
- undergo reproductive diapause = it is true but only in the case of adult diapause. If you speak here in general terms, just omit ''reproductive''.
- by all the orders of insects, such = many species in orders
- During dipause state, insects = Do not you see difference in diapause in general and adult (reproductive diapause)? If you write about reproductive development, then say that it is adult diapause. There is no reproductive development in egg or nymphaldiapause..
- events occur during the = that occur
- initiation of diapause can = again -- this is not about diapause in general, it is about ADULT diapause.
- or shape of the morphology or = there is no shape of morphology! You can speak about shape of some morphological structures.
- behavior also be recognized during the induction = English
- fates = fate
- insect after termination = only after or during and after?
Response
- We omitted this word which made confused to reader
- Now, we mentioned only the adverse conditions and removed the other factors
- These words were extra in that sentence and we removed those words
- Omitted “the”
- We removed facultative because we generalized the term
- As per editor’ advice, we changed the words as “many species in orders”
- We added the information about adult diapuse, so we wrote it as adult diapause
- As per editor’ advice, we added “that” occur
- We specified the sentence from general to adult diapause
- We checked English mistake and changed the verb
- We used “fate” inseated of fates
- We made a change and used “during” instead of “after”
Comments and Suggestions for Authors from Second paragraph
- with three generations in a year from = Where? In some region it produces 3, but in others - different number
- females produce the eggs depends on the availability of food and the reproductive males for mating process = English is poor.
- of the pedestris = omit
- instar = instars
- photoperiods [12] = 12 is about behaviour; Numata worked a lot on sensitivity
- The pedestris = omit
- pedestris insects entered = we do not say so
- photoperiod of the = at
- respond = response
Response
- We changed the sentence and added information about the environmental factor as “insect has more than one generation in a year depending upon the environmental conditions”
- We corrected the English and made this sentence more briefly
- Omitted
- We corrected instars instead of instar
- We added relevant reference
- We omitted “the” before species name
- We specified it to female adults instead of “insects”
- We corrected the grammar mistake and used “at”
- We corrected English mistake and changed it response
Comments and Suggestions for Authors from Reviewer 4 from Materials and Methods
Comments and Suggestions on First paragraph
- in a 40 cm3 cage under = a cage or many cages? 40cm3 is small (4*10*1 cm3) -- too small for 200 adults...
- 16hr light = --> h
- 8hr = --> h
- and water (0.05% ascorbic solution) = supplemented with
- harvested every
- Newly molted nymphs = ? hatched
- separate cages that were fed with water = cages were fed??? fed with wated? what is about food.
Response
- We used many cages for the rearing of insects
- We changed it as 16 h
- and 8 h
- as per editor’ advice, we added information as “supplemented with”
- we changed as collected every
- we used hatched instead of molted
- we specified the 1st instars nymphs in the sentence
Comments and Suggestions on second paragraph
- study the pedestris insects induced = adults?
- Begin ?
Response to
- We specified the stage as adult
we corrected the verb mistake
Comments and Suggestions for Authors from Reviewer 4 from Results
Comments and Suggestions on First paragraph
- inside of each
- (SD) and pre-reproductive (LD) females = justify
- observation (C). Ovary (O): Ovariole (OVA): Oocyte (OC): Lateral Oviduct (LOD): Common Oviduct (COD) and Spermatheca (SPE).
Response
- We corrected the grammar mistake
- We wrote the detail hours of long-day and short-day
- We used lowercase instead of uppercase alphabets
Comments and Suggestions on second paragraph
- Vitellarium
- vitellariums = plural - vitellaria
Response
- We used lowercase instead of uppercase alphabets
- We used singularform
Comments and Suggestions on third paragraph
- Diameter of Lateral Oviduct
- oviducts (fallopian tube) = this term is not used for insects!
- These fallopian tubes are
- to * p< 0.05, ** p < 0.01 = why bold?
Response
- We corrected the lowercase instead of uppercase alphabets
- We removed this term and used lateral oviduct
- We changed it to regular font instead of bold
Comments and Suggestions on fourth paragraph
- bug ( pedestris) = no need to present bith - common and Latin name here
- Accessory gland (AG), Vas deferens (VD), Testis (TES), Ejaculatory duct (ED)
Response
- We removed the brackets and Latin name
- We corrected the lowercase instead of uppercase alphabets
Comments and Suggestions on fifth paragraph
- Size of Accessory Gland
- clear was observed = ??
- Diameter of Ejaculatory Duct
Response
- We corrected the lowercase instead of uppercase alphabets
- It was overlooked and now added detail information as “a clear significant difference was”
- We corrected the lowercase instead of uppercase alphabets
Comments and Suggestions onSeventh paragraph
- Length and Width of Testes
- 10 independent biological replicates
- p< 0.05, ** p< 0.01.
Response
- We corrected the lowercase instead of uppercase alphabets
- We specified the adults instead of detail sentence from 10 adults.
- We changed it to regular font instead of bold
Comments and Suggestions for Authors from Reviewer 4 from Discussion
Comments and Suggestions on first paragraph
- ovary of pentatomid species = is anything known about alydids?
- conditions would be dangerous for their survival = not clear what you mean
- were based to distinguish = ?
Response
- We added detail information about Perillus bioculatus species which is family of true bug because there is much literature about Alydids to compare our results
- This sentence overlooked and now it have changed and made it meaningful according to the discussion
- We corrected the meaning of this sentence
Comments and Suggestions on second paragraph
- and investigated the
- reared on normal growth conditions.
- producing even
- the size
- males was poorly developed compare
- major programs take place during diapause
Response
- We added a clear sentence and made it clear now
- We added the non-diapause condition instead of “normal growth conditions”
- We corrected the sentence mistake in it
- We used development instead of size to make it clear the meaning of sentence
We changed the sentence style and added some information for further investigation
Comments and Suggestions for Authors from Reviewer 4 from conclusion
- reproductive insects of
- and develop as newly
Response
- We added adults instead of insects
- We corrected the sentence mistake in it
Comments and Suggestions for Authors from Reviewer 4 from conclusion
Response
We accepted all the suggestion and changed all the mistakes from the references
